# The Effects of Chinese Seafarers’ Job Demands on Turnover Intention: The Role of Fun at Work

**DOI:** 10.3390/ijerph17145247

**Published:** 2020-07-21

**Authors:** Yuan Gu, Dongbei Liu, Guoping Zheng, Chuanyong Yang, Zhen Dong, Eugene Y. J. Tee

**Affiliations:** 1College of Public Administration and Humanities, Dalian Maritime University, Dalian 116026, China; guyuan@dlmu.edu.cn (Y.G.); liudongbei@dlmu.edu.cn (D.L.); davidyang@dlmu.edu.cn (C.Y.); dongzhen@dlmu.edu.cn (Z.D.); 2Navigation College, Dalian Maritime University, Dalian 116026, China; captainzheng@dlmu.edu.cn; 3Department of Psychology, HELP University, Shah Alam 40150, Malaysia

**Keywords:** job demands–resources model, fun at work, occupational commitment, turnover intention, job demands

## Abstract

This study examines how an occupational commitment and a fun work environment serve as important mechanisms that influence the job demands–turnover intentions relationship. On the basis of the job demands–resources model, the study explored the relationship between job demands, occupational commitment, fun at work, and turnover intention. The hypotheses were (1) that job demands would be positively associated with predicted turnover intention; (2) that occupational commitment would mediate the job demands–turnover intention link and (3) that a fun environment would moderate the relationship between job demands and occupational commitment and between job demands and turnover intention. The study sampled 294 seafarers using an online survey, and applied descriptive, correlative analysis and the PROCESS Macro to test the hypotheses. Findings provide preliminary support for the three hypotheses, and contribute to a better understanding of the mechanism determining seafarers’ turnover intention. The results suggest the importance of holding appropriate group activities on-board to help seafarers alleviate fatigue and stress.

## 1. Introduction

Seafarers are an occupational group with one of the highest risks of stress [1]. Seafarers have to endure separation from family, burnout, time pressure, dealing with checks, workplace noise, sleep deprivation, cross-cultural communication, recreation scarcity, interpersonal stress, physiological needs, and heat due to job demands [2,3,4,5,6,7].

The high job demands and stressors seafarers are subject to create factors impacting not only on mental health (i.e., depression, anxiety, suicide, and alcohol or drug dependence), but also on turnover intentions [8,9]. Job demands may impact sleep deprivation, absence from society and home, and recreation scarcity and these can lead to fatigue, loneliness, and boredom. This explains why the number of junior cadets intending to pursue a career at sea continues to decline globally [2,10,11,12]. Studies reveal that a global shortfall of approximately 92,000 seafaring officers is expected by 2020 [13,14]. The shortage of skilled seafarers, particularly officers and engineers, has been a growing concern for the shipping industry. With more than 90% of world trade volumes transported by sea and extenuating factors such as the recent COVID-19 epidemic, this problem presents itself as both a global challenge and a concern for countries reliant on seafarers as part of their logistics operations. These contextual factors further emphasize the importance of understanding the psychological factors leading to turnover so that organizations can take the necessary steps toward in order to retain them in the workforce [13,14].

To date, several studies have been conducted with a focus on formulating strategies to retain seafarers. Most studies highlight the importance of improving mental health, physical health, well-being, job satisfaction, and stress relief, to retain seafarers [7,15,16]. However, little is known about the factors positively influencing turnover intention. Additionally, limited studies adopt a strong theoretical lens in explaining the relationship between job demands and seafarers’ turnover intentions.

The Job Demands–Resources (JD–R) model is developed based on Conservation of Resources theory [17] and is often used to explain the consequences of job demands [18]. According to this model, every occupation has its own specific risk factors associated with job-related stress, which can be classified into two general categories including job demands and job resources [19,20]. Job demands refer to physical, psychological, social, or organizational aspects of a job that require sustained physical and/or psychological effort or skills and are therefore associated with certain physiological and/or psychological costs [19]. These demands may include high work pressure, unfavorable or unsafe physical environments, and irregular working hours. Although job demands are not necessarily negative, they may turn into job stressors when meeting those demands requires high effort that exceed the resources that employees have [21]. For shipping safety and dealing with checks, seafarers have to stay focused during work and stay awake throughout the day and night. This further leads to fatigue and exhaustion. Along with an absence from society and home, seafarers increasingly feel a sense of prolonged isolation and loneliness [1,21,22]. Seafarers may thus work less effectively, consider the future to be hopeless, and have a stronger willingness to quit their job.

In this regard, fun at work is a relevant construct, with important implications for effectiveness across different organizational levels [23]. For example, a company may be considered a “fun” place to work in; thus, fun is positioned mainly as a feature of the work environment. Ford and his colleagues defined a fun environment as a work environment that “intentionally encourages, initiates, and supports a variety of enjoyable and pleasurable activities that positively impact the attitude and productivity of individuals and groups” [24]. This definition includes instrumental outcomes, such as an improved attitude and productivity, which are necessary for fun to gain momentum in organizations [25]. According to the JD–R model, a fun environment as a “replenisher” of psychological resources and not only buffers the impact of job demands on job strain, but also helps employees enhance their levels of control and helps them to recharge and recover from work demands faster [20,26,27,28,29]. Findings have shown that fun at work has a deep effect on organizations and employees [30,31]. For organizations, fun at work conduces to improving organizational effectiveness, for example, by increasing work passion, competitive advantage, elasticity [32,33], innovation, creativity [34], empowerment [35], applicant attraction [36], and productivity [33,37]. Empowered employees enjoyed their work, which encouraged more pleasure in their working roles and decreased their intention to leave [38]. For employees, findings have indicated that fun at work serves to enhance employee’s engagement [30,39], productivity, motivation [40], organizational citizenship behavior [39], and improves employee well-being [31,41], job satisfaction [42,43], commitment [44], and psychological capital [30]. At the same time, it has been shown to also relieve their anxiety, anger [36], emotional exhaustion [43], tardiness [45], and burnout [24,46], and to decrease their work pressure [33] and absenteeism, which eventually helps to reduce their intention to leave [24,36,46].

In light of the current shortage of officers and seafarers in the global shipping industry [8,9,10,47,48,49], turnover intention has become a crucial and topical concern. Ingersoll defined turnover intention as the likelihood that an employee would voluntarily leave an organization [50]. According to the JD–R model, turnover intention is one of the main outcomes caused by high job demands [26,27,28]. Work engagement [51], job satisfaction [51,52], organizational commitment [53], work–family relationship [54], and administrative leadership [55] have been found to be significant predictors of turnover intention.

For seafarers who have to face lots of demands such as shifting work hours, separation from home, and society and exposure to hazards [56,57], turnover intention may be an important associate of safety. For instance, Nielsen and his colleagues found that turnover intention was negatively associated with a motivation to follow safety procedures and that management prioritizes production over safety [58]. Relatedly, offshore workers, seafarers, and officers employed on cargo ships who perceived high levels of risk reported lower job satisfaction levels [15,59,60]. An association between job satisfaction and turnover intention is empirically supported [58,60,61]. In the maritime field, Kim and Lee also found that the more satisfied workers were over wages and working conditions, the lower the level of turnover intention of seafarers [61]. Although some other factors probably affect turnover intention as well, it is almost certain that a dissatisfied seafarer has the tendency to evaluate the cost of quitting with their decreasing occupational commitment and a search for alternative jobs [52].

Over the years, commitment has been defined and measured in many different ways [62,63]. This lack of consensus in the definition of commitment has contributed to its treatment as a multidimensional construct [62]. Occupational commitment refers to a psychological link between a person and his or her occupation that is based on an affective reaction to that occupation [62,64]. Meyer and Allen [62] identified three distinct themes in the definition of commitment, which are affective, continuance, and normative commitment. Meyer and his colleagues [64] developed measures of affective, continuance, and normative commitment to occupation with a sample of student nurses. Based on the above three-component model of occupational commitment, Li and Yan [65] developed a similar three-component model of occupational commitment including professional value, a sense of belonging, and professional efficacy. Professional value refers to an individual’s identification with his job, which will predict work engagement and other behaviors. Sense of belonging is conceptualized as the affection links between an individual and his occupation, and his organization. Professional efficacy refers to an individual’s perception about his competence as an employee. Moreover, Li and Yan developed a measurement of the three-component occupational commitment in the Chinese context with a sample of teachers. Occupational commitment is important because of its potential link to retention [64,66,67,68,69]. Employees with a strong occupational commitment will more strongly identify with, and experience, more positive feelings about their current jobs than will ones with a weak occupational commitment [64], and they have a lower tendency to leave their occupation [68,69,70].

Little was known about the antecedents of turnover intention and the role of fun at work in the maritime field. The current study fills the gaps in the literature by exploring several predictors of turnover intention that are of particular relevance to seafaring: job demands; fun at work; and occupational commitment.

**Hypothesis 1 (H1)**.
*Job demands would be positively associated with predicted turnover intention.*


**Hypothesis 2 (H2)**.
*Occupational commitment would mediate the job demands–turnover intention link.*


**Hypothesis 3 (H3)**.
*A fun environment would moderate the relationship between job demands and occupational commitment.*


**Hypothesis 4 (H4)**.
*Between-job demands and turnover intention.*


## 2. Materials and Methods

### 2.1. Sample and Data Collection

The approval of the research was received from Dalian Xinghang International Company and COSCO shipping Dalian branch, China. The cross-sectional data was collected on-board through online questionnaires.

Messages regarding research information were sent from the Crew Management Department of Dalian Xinghang International Company and the COSCO shipping Dalian branch to the captains, requesting them to notify seafarers on-board by clicking on the link to the online survey and finishing online survey forms at a time and place convenient to them. Each participant was told via the informed consent form that participation in this project was voluntary and that their responses will be kept in the strictest confidence and anonymity. The Crew Management Departments did not exclude or select any ship or seafarer when issuing the questionnaires. Although data on the number of seafarers on each ship informed of the research and asked to fill in the form were not available, the Crew Management Departments reported that if each seafarer they notified finished the survey, the sample size would be 380 in total (198 seafarers from Dalian Xinghang International Company, 182 seafarers from COSCO shipping Dalian branch). In fact, 294 valid questionnaires were collected. The effective questionnaire returns ratio is 77.4%. The sample of 294 respondents were merchant seafarers (148 seafarers from Dalian Xinghang International Company, 146 seafarers from COSCO shipping Dalian branch) and were predominantly captains (17.0%), male (87.0%), from rural areas (32.0%), with an average age of 34.01 (SD ± 9.978) and a junior college degree (41.5%). Most respondents earned less than 50,000 RMB per year (25.9%). The demographic characteristics of respondents are presented in Table 1.

### 2.2. Measurements

Job Demands: The study adopted the Job Demands Scale developed by Karasek [71], translated and revised by Zeng and Shi [72]. The scale included seven items, such as “your job requires you to work fast”. Responses are scored on a five-point Likert scale as follows: 1 = totally disagree and 5 = totally agree. In the present study, the Cronbach’s alpha of the scale is 0.88.

Fun Environment: The scale relating to a fun environment in the study was developed by McDowell [44] and included eight items, including ones such as “I get along well with my colleagues”. Responses were scored on a five-point Likert scale as follows: 1 = totally disagree and 5 = totally agree. In the present study, the Cronbach’s alpha of the scale is 0.87.

Occupational commitment was measured with the Chinese version of the Occupational Commitment Scale. Li and Yan developed a three-component occupational commitment in the Chinese context, and gave reliable and valid measurements of occupational commitment [65]. The scale is a 12-item scale measuring three subscales, which includes professional value (e.g., “I always take my work seriously and finish it on time”), a sense of belonging (e.g., “When introducing myself, I like to mention my occupation”), and professional efficacy (e.g., “I consider stress at work an opportunity rather than a threat”). Responses were scored on a five-point Likert scale as follows: 1 = totally disagree and 5 = totally agree. Li and Yan’s scale has a reported reliability of 0.947 (Cronbach’s alpha). In the present study, the Cronbach’s alpha of the scale is 0.86.

Turnover Intention: on the basis of the Scale of Turnover Intention developed by Mobley et al. [52], a scale of turnover intention including four items was designed including items such as “I often think about quitting my present job”, “I may leave the company for another position in six months”, “I plan to stay with the company for a long time”, and “If I continue with my present job, I will not have good prospects for development”. Responses were scored on a five-point Likert scale as follows: 1 = totally disagree and 5 = totally agree. In the present study, the Cronbach’s alpha of the scale is 0.73.

All of the scale reliabilities are presented in bold on Table 3.

### 2.3. Ethics Approval for Study

Ethics approval for this study was granted by the Science and Technology Department, Dalian Maritime University, Dalian, Liaoning, before data collection (3132020254). During data gathering, each participant was required to sign an Informed Consent Form. The Informed Consent Form included project overview, benefits, physical risks, confidentiality, freedom of participation, right of withdrawal, and communication of the results. Consent was obtained from all participants.

### 2.4. Preliminary Data Analysis

The current study used SPSS20.0 (IBM Corp. Released 2011. IBM SPSS Statistics for Windows, Version 20.0. IBM Corp., Armonk, NY, USA) to analyze the data through reliability and validity analysis, descriptive analysis, correlative analysis, and a PROCESS test (Model 8).

The present study conducted item analysis and normality tests to examine if there were any outliers from the data. The main purpose of item analysis is to find out the critical ratio (CR) value of each item in the questionnaire which determines whether each item is discriminative or not —and to delete the items that did not reach the significant level [73]. Item analyses were carried out separately for each scale and it was found that all coefficients turned out to be satisfactory (*p* < 0.05). No improvement was possible by eliminating items. During normality test, the skewness and kurtosis for latent variables were evaluated according to the procedure proposed by Kline [74]. None of these are problematic (see Table 2). Since these values are acceptable and normality corrections are usually only required for small samples [75], further sampling methods for normality were thus not completed.

## 3. Results

### 3.1. Descriptive and Correlative Analysis

The mean, SD, correlation coefficient, and the Cronbach α value are presented in Table 3. As expected, job demands were positively correlated with turnover intention significantly (r = 0.46, *p* = 0.000). Fun environment was positively related with occupational commitment (r = 0.68, *p* = 0.000), and negatively related with turnover intention (r = −0.14, *p* = 0.015). Surprisingly, job demands was positively correlated with occupational commitment (r = 0.39, *p* = 0.000). The result of the correlation analysis provides preliminary support for hypotheses verification in this study.

### 3.2. PROCESS Test

First, the present study applied Model 8 test in PROCESS to examine the theoretical hypothesis model, and found that job demands positively predicted turnover intention significantly (β = 0.206, 95% CI: (0.155, 0.256)). H1 was verified again. Although a fun environment does not moderate the effect of job demands on occupational commitment (β = −0.002, 95% CI: (−0.019, 0.014)), a fun environment moderates the effect of job demands on turnover intention (Effect Size = 0.008, β = 0.008, SE = 0.003, 95% CI: (0.003, 0.014)). H4 was supported, but H3 was not (details see Table 4). Further analysis demonstrated that the influence of job demands on turnover intention increases as the degree of a fun environment increases (details see Table 5). Specifically, before the two lines crossed, in the case of low job demands, the lower the degree of a fun environment, the higher the turnover intention. However, with the increase of job demands, although seafarers experience a high level of fun environment, their turnover intention was increasing, which may even be higher than that of seafarers with a low level of fun environment (see Figure 1). Furthermore, occupational commitment mediated the effect of job demands on turnover intention in Model 8 (β = −0.055, 95% CI: (−0.099, −0.010)), which verified H2—that occupational commitment acted as a part-mediator (see Table 4).

Therefore, the final model was presented as follows (see Figure 2).

## 4. Discussion

On the basis of the JD–R model, the present study explored the antecedents of seafarers’ turnover intention among 294 seafarers. H1, H2, and H4 were supported, and the testing of the hypotheses in this new context is one of the main contributions of this paper. The findings are elaborated below:(1)Job demands (including work stress, time pressure, and so on) positively predicted turnover intention. The result supported previous studies. Schaufeli and his colleagues [71,72,76,77], for instance, demonstrated that job demands (such as work stress, emotional demands) can result in an increase in turnover intention.(2)Occupational commitment partially mediated the relationship between job demands and turnover intention. This suggests that job demands not only predict turnover intention directly, but also affect turnover intention through occupational commitment. The result was consistent with previous research [64,66,67,68]. Surprisingly, job demands affect occupational commitment positively, contradicting the hypothesis. Further analysis shows that job demands was significantly positively correlated with professional value (r = 0.302, *p* = 000), sense of belonging (r = 0.270, *p* = 0.000), and professional efficacy (r = 0.437, *p* = 0.000). Although social recognition of seafarers continues to decline, seafarers are highly appreciative of their profession with high a level of work engagement. In addition, high job demands may represent their competence for their jobs. Moreover, seafarers are generally bored when they are not working due to the lack of recreational activities and the particularity of working onboard [1,2,4]. The job demands will keep them busy and make them feel like indispensable of the ship, so that their sense of belonging will be enhanced. What’s more, most of them considered job demands (including work pressure, time pressure, workload, and so on) as non-negotiable tasks. Gradually, they felt irreplaceable, and had more occupational commitment due to various reasons. Furthermore, seafarers generally have high identification with their jobs, and also expected greater social acceptance and recognition of seafarers.(3)A fun environment did not moderate the relationship between job demands and occupational commitment, which not only did not verify H3, but also was not consistent with the findings found by Ford, McLaughlin, and Newstrom [24]. The reason may be that the working conditions of the crew are highly unique because of closed environments and being at sea, a fun environment or not was not the key point to affect occupational commitment for seafarers.(4)A fun environment moderated the relationship between job demands and turnover intention. Specifically, in the case of low job demands, the higher the degree of a fun environment, the lower the turnover intention, supporting several previous studies [24,25,26,29,30,31,36,39,44]. However, with the increase of job demands, although seafarers experienced a high level of fun environment, their turnover intention was increasing, which may even be higher than that of seafarers with a low level of fun environment. The direction of the moderation effect is different from previous studies [78]. Accordingly, further communication with seafarers who participated in the survey is conducted, and show the reason may be that maritime field is different from others. In the case of low job demands, regarding the lack of recreation onboard, a fun environment could help increase seafarers’ job satisfaction and decrease their intention to leave to some extent, which is consistent with previous studies [24,25,26,29,30,31,36,39,44]. However, seafarers have to stay focused and awake throughout the day and night, and deal with some inspections during work [1,21]. When seafarers face high job demands, a high level of fun environment may shift from a kind of job resource to a kind of job demand because building and maintaining a fun environment required seafarers to make an effort. In other words, with a high level of job demands, seafarers may not have any extra energy to participate in recreational activities. Forcing them to join in the activities would increase their psychological burden and turnover intention. What they want is to finish their work as soon as possible and to take a good rest. Whereas, after a good rest, their willingness to create a fun environment would increase. Accordingly, in the maritime realm, the degree to which a fun environment is built depends on the level of job demands. Creating a fun environment and maintaining it appropriately may be a great benefit to organizations.

Seafarers’ job demands are high, and a number of seafarers choose to quit their jobs [10,11,12]. However, improving seafarers’ occupational commitment may indirectly reduce their turnover intention [62,63,64,67], suggesting that companies should provide a fun environment for seafarers to cushion the impact of job demands on turnover intention. At present, few recreational facilities and weak relationships with colleagues on ships reveal difficulties for crewmembers to feel a sense of belonging, and their enthusiasm to work will inevitably decrease. Therefore, shipping companies should create a fun environment for crewmembers according to their job demands to alleviate seafarers’ fatigue and stress. Since fun at work is defined by McDowell [44] as “engaging in activities not specifically related to the job that are enjoying, amusing, or playful.”, holding birthday parties, singing contests, sports meeting such as table tennis, running and other group activities on-board may contribute to a fun environment for seafarers. In addition, although certain shipping companies have political commissars on their ships, most lack psychological knowledge and an ability to solve the problems of crewmembers. Accordingly, political commissars would not have the ability to help others and themselves unless they receive more psychological training.

The present study explored several antecedents of turnover intention, but a few limitations exist. First, in terms of research sampling, all the subjects in this study were from Dalian Shipping Company, and the sample size was small; hence the generalizability is limited. Future research may investigate more seafarers from different regions to obtain more convincing results. Second, the present research was a cross-sectional study. Longitudinal research or experimental research can be used for future research to further explore the changing trends of crew turnover intention.

## 5. Conclusions

Based on the JD–R model, the present study explored the relationship between job demands, fun at work, occupational commitment, and turnover intention with a sample of 294 merchant seafarers. Findings indicated that (1) job demands could predict turnover intention positively; (2) occupational commitment mediated the job demands–turnover intention link, and (3) a fun environment would moderate the relationship between job demands and turnover intention. Specifically, the influence of job demands on turnover intention increases as the degree of a fun environment increases for seafarers.

## Figures and Tables

**Figure 1 ijerph-17-05247-f001:**
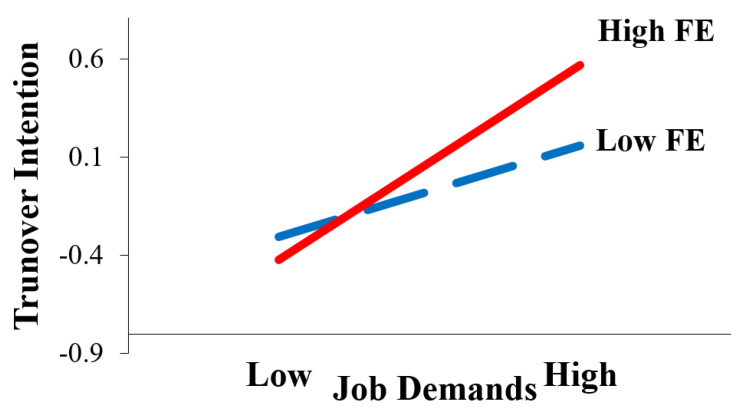
Interaction Plot. Note: FE = fun environment.

**Figure 2 ijerph-17-05247-f002:**
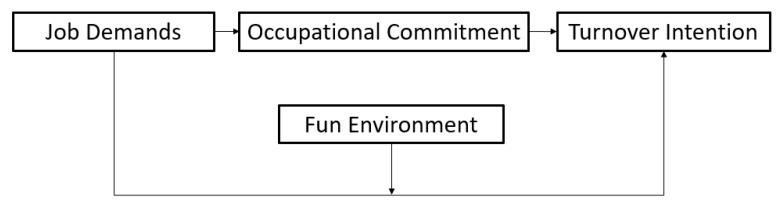
Final Model.

**Table 1 ijerph-17-05247-t001:** Sample Information (*N* = 294).

Demographic Variables		*N*	Percentage
Sex	Male	240	86.96%
	Female	36	13.04%
Hometown	Rural Area	94	85.46%
	Urban Area	16	14.54%
Age	Less than 20	6	2.21%
	20–29	92	33.82%
	30–39	96	35.29%
	40–49	66	24.26%
	50–59	12	4.41%
Education	Junior Middle School and Below	4	1.50%
	Secondary Technical School/Senior High School	40	15.04%
	Junior College	122	45.86%
	Bachelor’s Degree	98	36.84%
	Master’s Degree or Above	2	0.75%
Income	Less than 50,000 RMB	76	27.14%
	50,001–100,000 RMB	60	21.43%
	100,001–150,000 RMB	50	17.86%
	150,001–200,000 RMB	32	11.43%
	More than 200,000 RMB	62	22.14%
Jobs	Captain	50	18.12%
	Chief Engineer	12	4.35%
	Political Commissar	2	0.72%
	Chief Officer	32	11.59%
	Second Officer	12	4.35%
	Third Officer	26	9.42%
	First Engineer	4	1.45%
	Second Engineer	10	3.62%
	Third Engineer	14	5.07%
	Boatswain	2	0.72%
	Sailor	38	13.77%
	Carpenter	2	0.72%
	Machinist	16	5.80%
	Coppersmith	2	0.72%
	Master Mechanic	4	1.45%
	Chef	4	1.45%
	Waiter	44	15.94%

**Table 2 ijerph-17-05247-t002:** Results of the Normality Test.

	Job Demands	Occupational Commitment	A Fun Environment	Turnover Intention
Number of Items	7	12	8	4
Skewness	0.047	−0.337	−0.221	0.422
Kurtosis	0.212	1.449	0.130	0.421

**Table 3 ijerph-17-05247-t003:** Results of Descriptive and Correlation Analysis.

	M	SD	1	2	3	4	5	6	7	8	9	10
1. Job Demands	23.09	4.917	**(0.88)**									
2. Occupational Commitment	41.02	7.574	0.386 **	**(0.86)**								
3. Fun Environment	26.34	6.321	0.126 *	0.683 **	**(0.87)**							
4.Turnover Intention	11.61	2.059	0.462 **	−0.006	−0.141 *	**(0.73)**						
5. Sex	1.13	0.338	−0.094	0.071	0.077	−0.030	1					
6. Age	34.01	9.978	0.088	0.082	−0.058	−0.081	−0.463 **	1				
7. Education	3.20	0.756	0.056	0.091	0.037	−0.092	−0.164 **	−0.122 *	1			
8. Jobs	8.13	5.603	−0.258 **	−0.090	0.159 **	−0.059	0.584 **	−0.645 **	−0.298 **	1		
9. Hometown	1.64	0.483	0.194 **	0.128 *	0.071	−0.027	−0.273 **	0.567 **	−0.053 **	−0.422 *	1	
10. Income	2.80	1.509	0.035	0.130 *	0.035	−0.186 **	−0.382 **	0.566 **	0.263	−0.703 *	0.294	1 **

Note: * *p* < 0.05; ** *p* < 0.01. The reliabilities of the scales are presented in bold.

**Table 4 ijerph-17-05247-t004:** Results of Model 8 Test via PROCESS.

Dependent Variable	Independent Variable	Coefficient	Standard Error	t	*p*	LLCI	ULCI	R^2^
Occupational Commitment	Constant	37.966	3.560	10.665	0.000	30.953	44.979	0.530
	Job Demands	0.400	0.068	5.848	0.000	0.265	0.534	
	Fun Environment	0.722	0.052	13.996	0.000	0.620	0.823	
	Job Demands × Fun Environment	−0.002	0.008	−0.291	0.771	−0.019	0.014	
Turnover Intention	Constant	18.096	1.507	12.011	0.000	15.128	21.065	0.430
	Occupational commitment	−0.055	0.023	−2.422	0.016	−0.099	−0.010	
	Job Demands	0.206	0.025	8.081	0.000	0.155	0.256	
	Fun Environment	−0.030	0.024	−1.251	0.212	−0.078	0.017	
	Job Demands × Fun Environment	0.008	0.003	2.942	0.004	0.003	0.014	

LLCI and ULCI indicate the lowest and highest of the 95% Confidence Intervals respectively.

**Table 5 ijerph-17-05247-t005:** Results of Model 8 Test.

	Effect	Standard Error	t	*p*	LLCI	ULCI
Conditional direct effect(s) of X on Y at values of the moderator(s):						
A Fun Environment						
−6.121	0.154	0.031	5.052	0.000	0.094	0.214
0.000	0.206	0.025	8.081	0.000	0.155	0.256
6.121	0.257	0.031	8.231	0.000	0.196	0.319
Indirect effect of X on Y:						
Occupational Commitment	−0.026	0.012	0.990	0.000	−0.053	−0.005

LLCI and ULCI indicate the lowest and highest of the 95% Confidence Intervals respectively.

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
