# Peer review of "The Effects of Chinese Seafarers’ Job Demands on Turnover Intention: The Role of Fun at Work"

_ijerph, 2020, doi:10.3390/ijerph17145247_

Round 1

Reviewer 1 Report

Dear authors,

thank you for the possibility to review your article.

The study is interesting but the authors should improve several in the abstract, introdution, methods and result sections.

Line 11: At the beginning of the abstract, the authors should add a first phrase to introduce the topic of the study

Line 16: describe briefly the methods of the study

Introduction: Reduce the text and eliminate the subheadings 

Line 29: Eliminate the second "are"

Line 43: Eliminate the empty line

Line 131-133: Change 1, 2 and 3 with H1, H2 and H3

Line 134: Add H4 before "between"

Line 137-140: Move this description of Ethic aspects of the study in a specific paragraph in Materials and Methods Section

Line 141:  The authors should explain where, how and from whom the questionnaires were administered and what are the criteria of selection of the population in study

Line 145: Add ± after SD and before the number

Line 176: Before the results add two paragraphs, The first one with the description of Statistical analysis. A second one about the description of the approval of the study from Ethical Committee (line 137-140)

Line 230-276: Eliminate all the follow-up interviews

Line 290: What does it mean "provide a fun environment" for workers, especially for seafarers? The authors should explain briefly what they intent for fun environment.

Kind regards

Reviewer 2 Report

Thank you for the opportunity to review this manuscript. Here are my comments and suggestions:

INTRODUCTION: 

Line 43 and 44 could be on the same line in order to get the sentence to start with, 'Most...'

MATERIALS AND METHODS

Sample: How did the participants receive the online questionnaire?

How did you ensure anonymity and confidentiality? 

It would be best if you had a sub-section 'data instrument' and describe the questionnaire development and delivery. 

Was the questionnaire pilot-tested?

Data methods: need to put in the city of the SPSS version. 

RESULTS:

Following your test of internal consistency and reliability (cronbachs alpha test), did you remove any questionnaire items?

Were your cronbach's alpha values in your adapted questionnaires similar to the original questionnaires? 

DISCUSSION:

You have described interviews in your discussion. There was no mention of this in your methods. How did you select these participants? Where did you conduct the interviews? How did you analyse the interview data? All you present in the discussion is the transcript of the interview. What does that tell the readers?

OVERALL COMMENT

The manuscript provides good information but needs to be improved. Especially the interview part. It just appears without any analysis or discussion. 

Reviewer 3 Report

This paper applied the classic Job Demands–Resources model to a new context: seafarers. The paper is well-written. I really like the way the authors combine a quantitative study with a qualitative study. However, I have some concerns and suggestions for further improvement.

  1. Why the authors use PROCESS (based on OLS) to analyze the data? For survey studies, it is more common to use structural equation modeling. For this paper, I think using PLS-SEM is very appropriate. Maybe the author could justify why they use PROCESS?
  2. The authors could highlight the value of this study. So far, the most prominent value is applying the model to a new context. 
  3.  The authors wrote: "the influence of a fun environment on turnover intention increases as the degree of a fun environment increases". I think the authors mean: "the influence of job demand on turnover intention increases as the degree of a fun environment increases". 
  4. The direction of the moderating effect of the fun environment is very surprising. The positive effect of job demand on turnover intention is more prominent when the degree of the fun environment is higher. This seems to be contradictory to the authors' assumptions. Could you justify this issue? 
  5. I would suggest the authors show the full items as an appendix.

Reviewer 4 Report

The paper describes the relationship between job demands, fun at work, occupational commitment and turnover intention,and partly verified the previous hypotheses. The adopted methods appear scientifically sound to me. The research of the paper has certain social value.

I report several MINOR comments for the paper:

Line 15 and Line 307: Format needs to be unified.

Line 18:“Findings partly verified the three hypotheses” Maybe this sentence can be replaced with a clearer expression.

Lines 53: There are common sense mistakes in this line.

Line 148: This paper is mainly based on the data of unfamiliar people, so the author did not explain the reliability of the data, how to ensure that the investigators truly and objectively reflect their ideas? Can you give a brief supplement to the form of data collection?

Line 176: what is your" Model 8"?

Less...

Round 2

Reviewer 1 Report

Dear authors,

all my integration and correction requests have been met.

I suggest only to change the title at line 187 in "Statistical analysis" and to join the lines 192-203 under this paragraph, starting the results title at line 204.
Kind regards.

Author Response

Thanks for your comment. We have changed the title at line 187 to "Preliminary Data Analysis", starting the results at line 204.

Reviewer 2 Report

accept in current form

Thank you.

Author Response

Thank you very much

Reviewer 3 Report

I appreciate the efforts of the authors. However, the author did not address my concern regarding the direction of the moderation effect.

As you may see in the Figure, a fun environment strengthens the effect of job demand on turnover intention. This is not consistent with the authors' assumptions and implications. The direction is completely different. According to the findings, a fun environment is not recommended. The authors should address this issue carefully.
